# Nutritional Education Is an Effective Tool in Improving Beverage Assortment in Nurseries in Poland

**DOI:** 10.3390/healthcare9030274

**Published:** 2021-03-03

**Authors:** Anna Harton, Joanna Myszkowska-Ryciak

**Affiliations:** Department of Dietetics, Institute of Human Nutrition Sciences, Warsaw University of Life Sciences (WULS), 02-776 Warsaw, Poland

**Keywords:** water, beverages, nurseries, infants, toddlers, young children, nutritional education

## Abstract

The aim of the study was to assess the impact of education on improving the assortment of beverages served in nurseries in Poland. This analysis focused on beverages served to children with meals and between meals. We examined: water, tea, juices, soft/fruit drinks, compote, milk, cocoa, and coffee substitute. The study involved 93 nurseries enrolled in the Eating Healthy, Growing Healthy project and was conducted from 2015 to 2017. Nutritional education was carried out by dieticians or nutritionists and included 24 h of lectures given to the staff of nurseries. Training was conducted in a form of workshops that were divided into 12 topics. Additionally, consultations regarding infants and young children nutrition were available to the staff. Typically, education training was completed within 1–2 months in each nursery. The effectiveness of education was verified based on data on beverage supply in 186 ten-day menus, 1860 daily inventory reports before (at baseline), and 3–6 months after staff training (post baseline). Data were analyzed in a total group, and separately for public and non-public nurseries. Statistical analyses for categorical variables were done with Chi^2^ Pearson test. The differences were considered significant at *p* < 0.05. After education, a significant increase in the supply of water as a beverage served with meals was observed in all nurseries (68% vs. 87%; *p* = 0.002), both in public (72% vs. 90%, *p* = 0.036) and non-public (62% vs. 84%, *p* = 0.017) ones. Moreover, a decrease in served tea was noted: in all nurseries (71% vs. 44%, *p* < 0.001) and in public nurseries (77% vs. 42%, *p* < 0.001). Water was available between meals in almost all nurseries. Nutritional education for staff is an important tool for improving adherence to the nutritional recommendation in the case of core beverage supply in Polish nurseries.

## 1. Introduction

The period of the first three years of a child’s life is a time of very intensive growth and development. Nutrition of children at this age should be characterized not only by an adequate supply of calories and dietary nutritional value [1,2], but also by the correct assortment of consumed products and beverages [3,4,5]. According to the recommendation of the American Academy of Pediatrics (AAP) [6], the amount and quality of beverages consumed by a child is an important discriminant of the diet and an important component in the prevention of diet-related diseases. Results show that water consumption is associated with a higher quality diet, healthier nutritional habits, and lower risks of chronic disease [7]. On the other hand, there is a relationship between the consumption of sugar-sweetened beverages (SSBs) and an excessive weight gain in children [8]. Furthermore, a reduction in the consumption of SSBs by children has a positive effect on their body weight and health in general [8,9,10].

The healthy nutrition of a child is the responsibility of parents, who are models for their children [11], and teach them about food and nutrition. Shaping eating habits and food preferences take place very early in childhood [12]. Therefore, participation of parents in this process is crucial. The influence of parents on the child’s nutrition depends on many factors [13]. Parents’ education and their professional status might be reflected in the child’s eating practices [14]. Young children of working parents more often attend nurseries or kindergartens. In such an environment, a child spends many hours per day [15,16] receiving up to 70% of their daily nutrition [17]. For this reason day care centers must provide children with well-planned meals.

Planning a healthy diet for a young child is challenging—it is important to be familiar with the current recommendations [1,2,3,4,5] and effectively apply them in practice. This problem is further complicated by the frequent lack of nutritionists or dieticians employed by daycare centers in Poland. In nurseries without a dietician employed, other personnel (without a special nutritional training) are involved in menu planning [18]. The general character of Polish nutritional standards [1] makes people without adequate education insufficient to plan children’s nutrition. The existing situation necessitates constant education of the personnel, who after a special training might be able to plan a better menu for children. For those purposes, educational programs directed to various educational institutions, children of various ages, as well as their parents and caregivers, are effective. An example of such program is the WIC (Nutrition Program for Women, Infants, and Children) [19], which was implemented for many years in the United States. Furthermore, such activities are carried out in primary schools located in the Hunter New England region of New South Wales, Australia [20], and in kindergartens of many European countries, and also in Poland as the ToyBox project [21,22].

Nutrition education was the basis of our program Eating Healthy, Growing Healthy (EHGH) [23]. The uniqueness of this program was that it was comprehensive, addressed to children and their staff at the nurseries, and indirectly to their parents. Our previous research on kindergartens [24,25], also carried out within the EHGH project, has shown an improvement in the quality of children’s nutrition (e.g., increasing the supply of fresh vegetables and fruits, milk and dairy fermented beverages) and a favorable change in the assortments of beverages after staff education. To date, there has been no comprehensive educational program in Poland addressed directly to nurseries. Therefore, undertaking such a program and research was necessary. In Poland, nurseries are attended by infants (after reaching the age of 20 weeks) and young children up to three years old. Hence, young children are dependent on caregivers and their knowledge. The research carried out in a representative group of caregivers showed that their nutritional knowledge is not satisfactory [26]. Additionally, there is no research in Poland assessing the effectiveness of educational activities. Project EHGH is the answer to amending that lack, including the nurseries. There are two types of nurseries in Poland: public and non-public. In this context, it was an interesting aspect: what is the impact of the type of nursery on children’s nutrition? The public and non-public nurseries in Poland differ in the method of financing. Public nurseries are financed by public money/by the government. In the non-public ones, the entire childcare charge is paid by the parents, and is only sometimes subsidized by the government. This situation may be reflected in the number of enrolled children, the organization of nutrition (own kitchen/catering), and the amount of money spent on the nutrition of the children. The data obtained during the EHGH project allows us to perform such analyses. In addition, we could analyze these data in conjunction with the assessment of the effectiveness of nutritional education. To the best of our knowledge it has not been a subject of any research so far.

The data presented in this article are a part of the EHGH program. The aim of this study was to assess the impact of staff education on improving the assortment of beverages served in nurseries in Poland. Data were also analyzed with respect to the type of nurseries: public vs. non-public.

## 2. Materials and Methods

### 2.1. General Information about the EHGH Project

The EHGH project was dedicated to nurseries and kindergartens from across Poland. It was implemented from January of 2015 to December 2017 and funded by the Danone Ecosystem Fund. The purpose of the project was to improve the quality of menus in day-care centers (DCCs) by providing nutrition education to staff. It was conducted directly in person (by specially trained educators) and online (access to educational materials on a website, indirect participation). The overview of the EHGH project is presented as Appendix A and has been described in previous articles [18,24,25,27,28]. All activities performed in the project were free of charge for DCCs and participation in the project was voluntary. As many as 2638 DCCs participated in the EHGH project, with the directed education covering 1347 DCCs and 13,214 employees of these institutions. The majority of the total number of participating institutions were kindergartens, because their share in the total number of DCCs in Poland is much larger than nurseries. Nurseries cover only 10% of needs in this area [28,29].

### 2.2. Study Participants

The scheme for selecting the samples of the present study is presented in Figure 1.

Due to the criteria for inclusion, results only from directly participating nurseries (public and non-public) are considered. The inclusion criteria for nurseries were: cooperation of the center with an educator from EHGH, providing of ten-day menus and inventory documents, working full-time (more than 5 h a day), providing full-board nutrition (defined as more than two main meals (breakfast, lunch) and one snack).

The next step was to select nurseries with access to drinking tap water. Drinking water, also known as potable water, is water that is safe to drink. According to the World Health Organization’s 2017 report, safe drinking water is water that does not represent any significant risk to health over a lifetime of consumption, including different sensitivities that may occur between stages of life [30]. Drinking water quality standards describe the quality parameters set for drinking water; in Europe, this includes the European Drinking Water Directive [31]; 128 nurseries met these criteria. These nurseries were attended by 8182 children under the age of 3 years. The 8182 children attending the examined nurseries constituted 10% of all children enrolled by care services for infant and young children in Poland in 2016 [29].

The next inclusion criteria was completion of the entire planned direct nutrition education program, which included 24 h of lectures conducted in a form of workshops that were divided into 12 topics and additional consultations on the subject of nutrition for infants and young children. The nutritional education was prepared by experts such as dieticians and nutritionists as well as behavioral scientists and healthcare practitioners. The nutritional education was conducted by 180 trained educators. The educators were people with higher education, who completed their studies in the field of dietetics or human nutrition. These people were specially recruited for the EHGH program and underwent 6 months of training before the program began. The whole course was completed with an exam, verification of acquired knowledge, and obtaining the EHGH Educator certificate.

The curriculum of the 12 topics included: general recommendations of balanced nutrition; water and its role in children’s diet; sugar in children diet; prevention of food allergies; fussy eaters or food neophobic; strong bones and teeth—the role of vitamin D and calcium; salt in children’s diets; attractive presentation of the meals; plump, overweight, obese—how to recognize the problem; limits of child’s choice; involving kids in cooking; serving size-self-eating, self-deciding. The last three classes concerned psychological aspect. Moreover, topics included problems such as: who makes decisions on the choice of products for a child, encouraging—yes or no to independent choices, and where and what are the limits of child’s choices (child’s role vs. caregiver’s role)? This topic underlined the fact that children can also take decisions about their meals; it is a simple way to educate them in nutrition. Surveys regarding the satisfaction of nurseries from participating in the EHGH project showed that the topic about water and its role in children’s diet drew great interest [26]. Before education, 128 nurseries participated in the research. The criteria of inclusion (completed education program) were finally met by 93 nurseries. Excluding criteria were: lack of the required number of 24 h of education, incompletion of all topics on the nutrition program, as well as lack of cooperation with the educator in the scope of transferring the necessary documentation (ten-day menus and daily inventory reports before and after education). Despite the high rating of the program that was shown in the evaluation after the completion of all activities in 2017 [26], 27% nurseries were excluded from this study due to failure to fulfill the criteria described above.

### 2.3. Evaluation of the Nutritional Education Based on Analysis of Beverages Supply

Evaluation of the nutritional education was based on the assortment of beverages served in 93 nurseries. The assessment was based on ten-day menus (10 consecutive days) and daily inventory documents. The inventory reports included all food products used in the kitchen to prepare all meals and beverages for children. The example of a daily inventory report is included in another publication as a Appendix A [24]. Ten-day menus and daily inventory reports were used in both comparative analyses: assessment I—before education, and assessment II—3–6 months after education. Typically, education training was completed within 1–2 months in each individual nursery. A total of 186 ten-day menus and 1860 daily inventory reports were analyzed during this study. Evaluation of the nutritional education based on an analysis of beverage supply in nurseries is presented in Figure 2.

This analysis focused on the assortment (defined as presence/absence) of beverages served with meals or available to children between meals. We took into account beverages served with meals, such as water, tea (regular/black), herbal/fruit tea, juices (100% natural), soft/fruit drink, compote (compote is a beverage made of typical Polish fruits as apples, strawberries, cherries, currants boiled in large amount of water with added sugar), milk, cocoa, and coffee substitute (usually wheat; made from roasted barley, beetroot and chicory), as well as water between meals. Recipes of the traditional Polish beverages and their average nutritional value are included in another publication [27]. Based on the nurseries’ menus, we checked if the beverages were present at least once a week (five consecutive days). This rule did not apply to water served between meals—water had to be available to children every day, otherwise it was not counted. Data compliance was verified by their occurrence in inventory reports, in this case the same rules as in the menu were followed. The collected data were expressed as the number and percentage of nurseries suppling some kind of beverages. The study did not intend to measure the quantity of each beverage offered or consumed; the portion size will be discussed in another study.

### 2.4. Statistical Analysis

All data were processed statistically using Statistica version 13.1 (Copyright©StatSoft, Inc., 1984–2014, Cracow, Poland). Means, standard deviations (SD), medians, minima, and maxima were calculated for the continuous data such as: total number of children attending nurseries including children < 1 year of age or the financial fee (applies only to nutrition fees—it is the cost of purchasing food products). For the categorical data such as type of catering* (definition below) only numbers and percentages of the participating nurseries are presented. All data were analyzed across the total group and by the type of nursery (public vs. non-public). Statistical significance for categorical variables was determined by the Chi^2^ Pearson test. In the case of continuous variables, the Shapiro–Wilk statistic test for testing the normality was used. Due to the lack of normal distribution of continuous data, the U Mann–Whitney test was used to check for significant differences. The differences were considered significant at *p* < 0.05.

*Type of catering—a way of organizing nutrition for children. It depends on the type of nursery and catering facilities, including the possession of a kitchen and employment of kitchen staff. Defining and characterizing subcategories for statistics:

Own kitchen—meals are prepared for children in the nursery from scratch.

Internal catering—meals are prepared by the kitchen of one nursery, and are distributed to others.

External catering—meals are purchased from an external company.

Mixed (own kitchen and external catering)—selected meals are prepared by the nursery from scratch (usually breakfast and afternoon tea), and others (e.g., lunch/dinner) are purchased from external catering.

## 3. Results

### 3.1. General Characteristics of DCCs

The study involved 93 nurseries from all over Poland, attended by 6.099 children under the age of 3; infants accounted for only 5% of the total group. Table 1 presents the general characteristics of the total number of nurseries and the type of nursery (public vs. non-public nurseries).

Significant differences between types of nurseries were noted in terms of the number of children, type of catering, and financial fee. The public nurseries compared to the non-public ones were characterized by a higher average number of attended children, more often they run own kitchen, but they also had lower fees for child nutrition.

### 3.2. Beverages Served in Nurseries Before and After Nutritional Education

The assortments of beverages offered to children in nurseries both before and after nutritional education for the total group of nurseries and different types of nurseries are presented in Table 2.

Considering the total number of nurseries after education, the percentage of nurseries providing water increased by 19% and supplying tea (regular/black) decreased by 27%. The compote and milk remained the beverages served most often to children with meals. Considering the type of nursery, in public nurseries the same significant changes were noted as for the total number of nurseries. In the case of non-public nurseries, the significant differences were seen only for the serving of water. The slight reduction observed in the number of nurseries offering children tea to drink was not statistically significant (*p* ≥ 0.05).

Additionally, analyses were conducted to compare the assortment of served beverages in public vs. non-public nurseries before and after education (Table 2). Before education, a smaller number of public nurseries (91%), compared to non-public ones (100%), offered children water between meals, with a higher number of nurseries serving juice with meals (respectively: 52% and 16%). After education, a large number of public nurseries, compared to non-public ones, still offered compote (respectively: 98% and 87%), cocoa, and coffee substitute (respectively: 90% and 71%) as well as juice (respectively: 69% and 16%). Data are presented in Table 2.

## 4. Discussion

Beverages are an important part of a child’s diet and a significant element in the prevention of many diet-related diseases. Their assortment should be chosen according to the nutritional recommendations and adjusted to the child’s age [1,2,3,4,5]. Diet planning for young children requires knowledge in this area. Caregivers do not always have sufficient knowledge. Nutritional education is a way to improve nutritional knowledge, which can translate into better planning of a child’s diet. However, education should be evaluated in terms of the effectiveness of the undertaken actions, which was the purpose of this study in relation to beverages served in nurseries.

Before education in nurseries, water was rarely served with meals. However, it is worth noting that in most nurseries (96%) water was available to children to drink between the meals. Another study [27,32] reported a slightly smaller percentage of DCCs offering water between meals to children. The authors reported that they included water with honey or fruit syrup as well. According to the recommendations for infants and young children [1,4,5] pure water should be the first choice to quench thirst. In the past decades, water was not a popular drink in Poland because it was associated with poorer nutritional value compared to other drinks. In recent years, however, the popularity of water has increased significantly. The current situation can be due to the educational campaign “Mom, Dad, I prefer water!” also targeted at children and their caregivers, but it is directed at kindergartens and schools, not nurseries. Our educational program was also aimed at popularizing water consumption. As the personnel of nurseries indicated, “Water and its role in children’s diet” was a very popular lecture [33]. In Poland, water was introduced in the infant feeding scheme for the first time in 2014, and since then its popularity has been growing. However, it is not always offered to children in accordance with nutritional recommendations [26]. As demonstrated in the current study, despite the availability of drinking water between meals before education, it was not the first choice. This shows that availability does not translate into supply without adequate nutritional knowledge and awareness of staff. In addition, the lack of supply translates into a lack of consumption by children, and this definitely affects the development of eating habits in early childhood [12]. It has been shown that nutritional education can be effective because the supply of water with meals was significantly increased in all nurseries after staff education. Furthermore, another study conducted in the same EHGH project in 478 kindergartens in Poland showed an increase in the share of water in child nutrition as a result of nutritional education [25].

Tea was another drink for which supply has changed as a result of education. In this case a decrease in supply was recorded. Tea is not a drink recommended for children due to its composition. This kind of beverage contains the most active ingredients such as polyphenols or caffeine. In Poland, even infants and young children are served tea [18]. Authors of another study investigating a representative group of Polish infants and young children demonstrated that tea is introduced into their diets [34]. Tea was regularly offered to 5% of infants at the age of 4 months. A relatively low share of tea compared to other drinks was reported in studies of children aged 1–5 years from different European countries [35].

Before education in nurseries, the most popular drink served with meals was compote. A study conducted in preschools also showed that compote, not water, was given to children most often [27]. In the current study, nutritional education has not changed supply of the compote. This traditional Polish drink is an example of sugar-sweetened beverage often served with dinner. This tradition also dominates in homes, hence even young children know this drink. In our opinion, the lack of effect of nutritional education in reducing this kind of drink in the diet of children was caused by the attachment to Polish cuisine. Another reason for the inefficiency of education could be caused by children’s preferences. Sweet taste preference, which is present in children, is also present in adults. Hence, people planning nutrition in nurseries, as well as chefs, could be following their own taste preferences. The lack of effectiveness of education in the supply of SSBs may be a result of low awareness of the staff about the nutritional values of beverages. People involved in feeding children without proper nutritional education are often guided by their experience and they often acquire their knowledge from the Internet [36]. Therefore, to improve the quality of the nutrition of young children and increase the effectiveness of all educational activities, training in this subject should be included in the state policies. The first step should be the introduction of mandatory employment of a dietician in the nurseries. Other Polish studies proved to have a greater effectiveness of implemented nutrition education in nurseries in which dieticians were employed [32] compared to nurseries without them. Results from our study show that even if an effort is made to educate people without prior substantive preparation, the benefits they receive may not be completely satisfactory. The increase in the effectiveness of nutritional education could have been more significant if qualitative nutrition recommendations would be obligatory for nurseries in Poland. Such mandatory recommendations exist for kindergartens and schools; they were introduced in Poland in 2015. These regulations applied to the assortments of drinks, including the sugar share. The legitimacy of such laws has already been demonstrated in preschool studies [37]. However, the authors emphasize that education should be run in parallel with the law. According to the Polish law, nurseries only have to comply with Polish nutritional standards [1]. However, these standards do not specify the product range, only the supply of calories and individual diet components.

No change in the supply of SSBs after education of the staff is directly translated into their planning of child nutrition, and eventually higher sugar consumption by children. The World Health Organization (WHO) [38] is paying special attention to dietary sugar content, and studies highlight the fact that the increasing intake of sugar in the form of free sugars is directly correlated with the consumption of SSBs [7]. A high sugar content in beverages leads to a risk of dental problems in young children [39] and an increase in their body mass [40]. A positive correlation between consumption of SSBs and BMI (Body Mass Index) in children aged 2–5 is also indicated by DeBoer et al. [41] Childhood obesity is a public health problem in many countries [42], also in Poland [43], especially in the group including the youngest children [34]. WHO experts [38] emphasize that free sugars contribute to the overall energy density of diets, and may promote a positive energy balance. This is correlated with obesity, type 2 diabetes, cancer, and cardiovascular diseases, and is inversely correlated with a healthy diet [7,44,45,46,47]. Recognizing the importance of the early prevention of diet-related diseases and adhering to the AAP recommendations [5] in subsequent educational activities, special attention should be paid to reducing the supply of SSBs in nurseries. Such a reduction in the supply of SSBs under the influence of staff education was noted in our other research carried out in kindergartens [25]. In the terms of general considerations regarding the effectiveness of education, it is worth taking into account the type of nurseries to which the activities are addressed. In our study, public and non-public nurseries differed in the number of children, type of catering and the financial fee per child/day. Such differences can determine not only the staff employed but also the children’s nutrition. Before the education, a smaller number of public nurseries compared to non-public ones, made water available to children between meals or more frequency served juice (100% natural) with meals. More frequent serving of juices and SSBs in public nurseries was also noted in other studies [18,32]. After the training, a larger number of public nurseries, compared to non-public ones, still offered compote and juice. The lower financial fee, which is more characteristic of public nurseries, does not have to be related to a worse menu of children [18]. However, in this case, public nurseries gave children unrecommended SSBs more often. Additionally, in Poland, the SSBs and juices are more expensive than water, which also contradicts the nurseries’ finances; public nurseries usually have a lower income compared to non-public ones. On the other hand, having their own kitchen gives the nurseries greater opportunity to implement the recommendations in practice. This should also be reflected by the appropriate variety of beverages offered to children. In our study, having their own kitchen was associated with more frequent preparation of compote for children while it is not typical for catering, which usually supplies commercial beverages such as SSBs. Due to the differences coming from the type of nursery, it is worth considering the type of nurseries when assessing the effectiveness of the undertaken activities. General child nutrition practices should be looked at before staff training. Further education is needed to improve knowledge and awareness on the subject. It is very important to show the relationship between the consumption of SSBs and the health of children.

In planning nutrition education, one should take into account all individuals engaged in the child’s nutrition. The role of parents and the family environment is crucial here. Children identify with family members and adopt all patterns, views, and eating habits from them. The relationship between the children, parents, and other caregivers in nutritional behavior has been raised in many studies [48,49,50].

### Strengths and Limitations of the Study

The presented research has several strengths and some limitations. A major strength of this study was the large sample of nurseries from across Poland—the majority of surveys available in Poland cover only individual nurseries from selected regions of the country. Moreover, the study included both public and non-public institutions. The method of data collection, a large number of documents (ten-day menus from 10 consecutive days and daily inventory reports), acquired from nurseries was also unique. Our research had some limitations. Although the study covered a large number of nurseries (public and non-public) from all over the country, these were not representative of all nurseries in Poland due to our inclusion methods (DCCs enrolled to the program). The introduced changes were assessed only once; it would be beneficial to repeat the assessment after a longer time period (e.g., 1 year). Another limitation of the conducted study was the lack of a control group (a group without intervention/nutritional education), which may to some extent limit the final conclusions.

## 5. Conclusions

Nutritional education for staff is an important tool for improving compliance to the nutritional recommendation in the case of water and tea. After education, a significant increase in water and decrease in tea servings with meals to children in all nurseries were noted. Water was available between meals in almost all nurseries. However, in the nutrition education plan, it is worth taking into account the type of institution (public vs. non-public), because it is connected with the number of children, the type of catering, and the financial fee, which determine the planning of children’s nutrition. Well-planned nutrition with the proper choice of beverages in a nursery has crucial importance for healthy growth and development of children. There is a constant need for continuing education for people responsible for infants’ and toddlers’ nutrition in Poland. To increase its effectiveness, there is a need to develop practical recommendations and guidelines to implement in DCCs.

## Figures and Tables

**Figure 1 healthcare-09-00274-f001:**
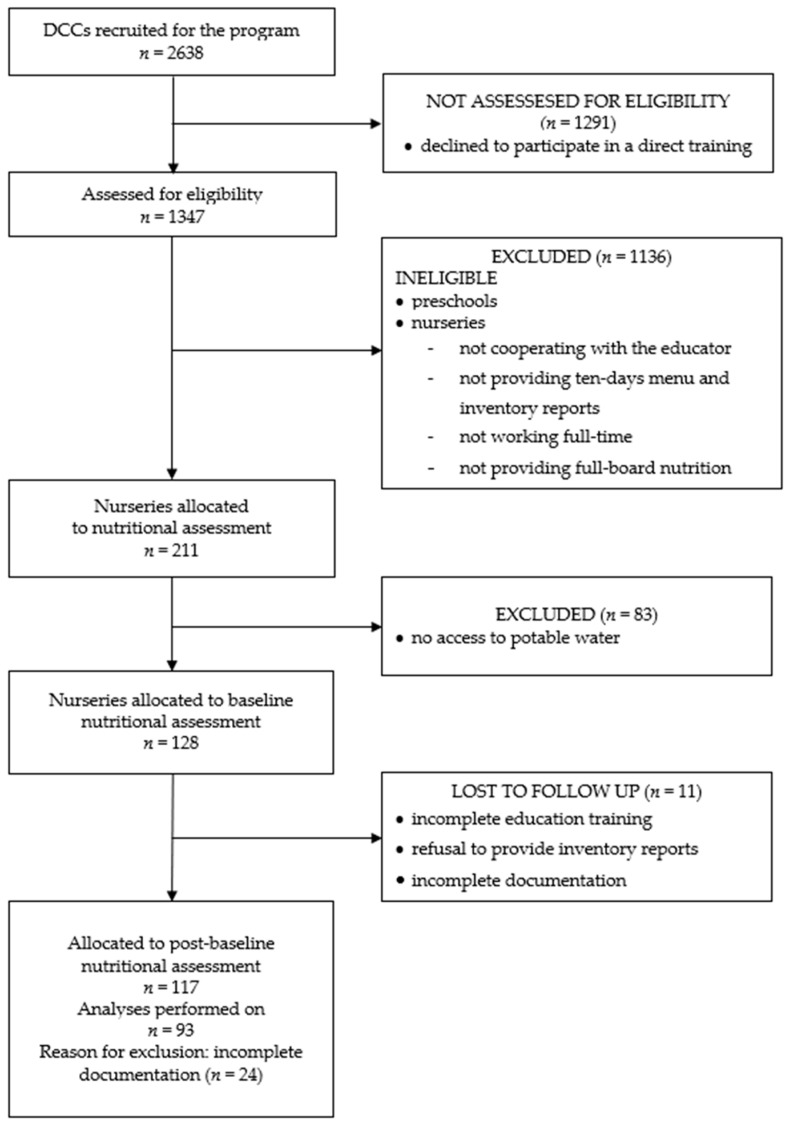
A scheme for selecting the sample of the study.

**Figure 2 healthcare-09-00274-f002:**
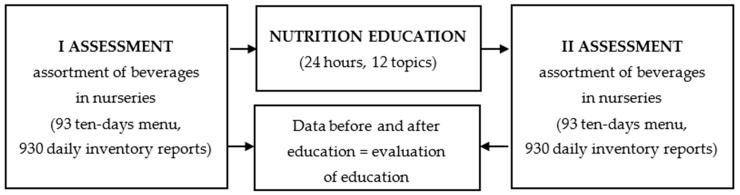
Evaluation of the nutritional education based on analysis of beverages supply in nurseries.

**Table 1 healthcare-09-00274-t001:** General characteristics of nurseries.

Category	Total Number of Nurseries	Public Nurseries	Non-Public Nurseries	*p*-Value
	(*n* = 93)	(*n* = 48)	(*n* = 45)	
Data on children attending nurseries				
Total number of children:				
N/%	6099/100	4566/75	1533/25	<0.001 *
Mean ± SD	66 ± 1.3	95 ± 47.4	34 ± 24.1	
Median; Min—Max	53; 5–180	94; 20–180	26; 5–114	
Children < 1 year of age:				
N/%	277/5	171/4	106/7	0.0501
Mean ± SD	3 ± 6.4	4 ± 7.2	3 ± 5.4	
Median; Min—Max	1; 0–43	2; 0–43	1; 0–25	
Data on nurseries				
Type of catering ^(a)^:				
Own kitchen: *n*/%	53/57	44/92	9/20	<0.001 **
Internal catering: *n*/%	3/3	1/2	2/4.5	
External catering: *n*/%	2/2	0/0	2/4.5	
Mixed (own kitchen and external catering): *n*/%	35/38	3/6	32/71	
Financial fee ^(b)^:				
Per 1 child/day/PLN ^(c)^				
Mean ± SD	7.7 ± 3.1	5.4 ± 1.6	10.1 ± 2.4	<0.001 *
Median; Min—Max	6; 4–15	5; 4–12	10; 5–15	

SD—standard deviation; N/%—number/percentage of children; *n*/%—number/percentage of nurseries; ^(a)^ way of preparing meals for children, explanation of the sentence and categories in Section 2.4, ^(b)^ applies to the cost of purchasing food products; ^(c)^ PLN (Polish New Zloty) = ~0.23 Euro; * *p* < 0.05—significant differences between type of nurseries (U Mann-Whitney test: significant difference *p <* 0.05); ** *p* < 0.05—significant differences between type of nurseries (Chi^2^ Pearson test, *χ*^2^—56.24, df = 3); *p* ≥ 0.05—no significant difference.

**Table 2 healthcare-09-00274-t002:** Assortment of beverages offered to children in nurseries before and after nutritional education for the total group as well as considering the type of nursery.

Beverages Offered in Nurseries	Total Number of Nurseries (*n* = 93)			Public Nurseries (*n* = 48)			Non-Public Nurseries (*n* = 45)		
	Education			Education			Education		
	Before *n* (%) of Nurseries	After *n* (%) of Nurseries	*p*-Value	Before *n* (%) of Nurseries	After *n* (%) of Nurseries	*p*-Value	Before *n* (%) of Nurseries	After *n* (%) of Nurseries	*p*-Value
Water available between meals	89 (96)	91 (98)	0.406	44 (91) *^1^	47 (98)	0.168	45 (100)	44 (98)	0.315
Compote	84 (90)	86 (92)	0.601	45 (94)	47 (98) *^2^	0.307	39 (87)	39 (87)	1
Milk	84 (90)	79 (85)	0.791	46 (96)	41 (85)	0.168	38 (84)	38 (84)	1
Herbal/fruit tea	75 (81)	80 (86)	0.325	38 (79)	40 (83)	0.601	37 (82)	40 (89)	0.368
Cocoa and coffee substitute	69 (74)	75 (81)	0.292	39 (81)	43 (90) *^2^	0.247	30 (67)	32 (71)	0.648
Tea (regular/black)	66 (71)	41 (44)	<0.001 *	37 (77)	20 (42)	<0.001 *	29 (64)	21 (47)	0.089
Water	63 (68)	81 (87)	0.002 *	35 (72)	43 (90)	0.036 *	28 (62)	38 (84)	0.017 *
Juice (100% natural)	32 (34)	40 (43)	0.228	25 (52) *^1^	33 (69) *^2^	0.094	7 (16)	7 (16)	1
Soft/fruit drink	18 (19)	10 (11)	0.1	12 (25)	5 (10)	0.061	6 (13)	5 (11)	0.748

*n*—number of nurseries, * Chi^2^ Pearson test (*p* < 0.05)—significant differences before and after education separately: in total nurseries (Tea *χ*^2^—13.75, df = 1; Water *χ*^2^—9.96, df = 1), in public nurseries (Tea *χ*^2^—12.48, df = 1; Water *χ*^2^—4.37, df = 1), in non-public nurseries (Water *χ*^2^—5.68, df = 1); *^1^ Chi^2^ Pearson test (*p* < 0.05)—significant differences before education in public nurseries vs. in non-public nurseries (Water available between meals *p* = 0.047, *χ*^2^—3.91, df = 1; Juice (100% natural) *p* < 0.001, *χ*^2^—13.73, df = 1); *^2^ Chi^2^ Pearson test (*p* < 0.05)—significant differences after education in public nurseries vs. in non-public nurseries (Compote *p* = 0.039, *χ*^2^—4.22, df = 1; Cocoa and coffee substitute *p* = 0.024, *χ*^2^—5.07, df = 1; Juice (100% natural) *p* < 0.001, *χ*^2^—26.81, df = 1); *p* ≥ 0.05—no significant difference.

## Data Availability

The database for the current study are available from the corresponding author on reasonable request.

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
