# Peer review of "Nutritional Education Is an Effective Tool in Improving Beverage Assortment in Nurseries in Poland"

_healthcare, 2021, doi:10.3390/healthcare9030274_

Round 1

Reviewer 1 Report

The article, “Nutritional education is an effective tool in improving water supply in nurseries in Poland: A nationwide cross-sectional survey”, by Anna Harton and Joanna Myszkowska-Ryciak seeks to assess the impact of education on improving the assortment of beverages served in nurseries in Poland. The authors concluded that “nutrition education of staff is an important tool for changing beverages assortment in nurseries”.

It is a valuable research and the manuscript has been written with a clear flow of headings and informative explanation of each section, though some details have not been given in the ‘right place’ (further comments below).

The authors’ suggestion towards making the nutritional educations obligatory for nurseries in Poland, has been well supported through this manuscript. However, enhancing the nutrition knowledge and its relationship with positive long-term effectiveness on children’s food pattern and eating behaviours plus lifelong health benefits is not a novel finding. But perhaps findings of this study can highlight the need for action(s) made by Polish policy makers, who are involved in legitimacy of such law.

Below are some suggested amendments and questions for the authors:

  • The abstract needs to be reviewed for considering more clarifications e.g. line 12 “...served with meals and between meals” considering adding ‘to children’.
  •  

Line 26- the conclusion stated in abstract must be consistent with the manuscript conclusion.

  • Line 135- for Figure 1 some details related to exclusion criteria e.g. full-board nutrition must be added. Generally, figures and tables must stand independent from the body of the draft.

  • There are several times that authors used “small children” instead of “young” or “toddler” or infant- this must be amended through the whole manuscript with a consistency. For example, line 151 or line 327.

  • There are few grammar errors, such as line 174- a more careful proofread is highly advised.

  • I wonder why the number of ten-day menus and daily inventory reports has been stated differently through the body (line 194) and Figure 2 (line 199)?

  • Though method section has been well written but there are different items which have not been addressed for their collection. Instead they had been referred in other sections like results or discussion sections. Examples are financial fee, types of nurseries (the definition of public and non-public has been given through the discussion, line 383), the age of participants (under 3yrs- stated in results line 233) and type of catering.

  • Line 236- Table 1, there is a confusion in reporting the median; min-max age- 1 (year?)- 0-43 (months?)- this must be corrected and reported in either years or months.

  • Line 232- authors stated that children under the age of 3 have been recruited but the reported mean and median for age in Table 1 for public nurseries reflecting older than that age. How this can be clarified better in defined age for recruited children?

  • Line 236- Table 1, more information such as df and χ2 has been missed for the done Chi2 test to reveal the differences between type of nurseries.

  • Line 308- the sentence needs to be supported by a suitable reference, because nurseries feedback on given lectures through the intervention (I mean the given quote, line 309) is not related to the aim of this study either its findings.

  • Line 317- lack of suitable reference. Statements must be evidence- based rather than general ones without citation(s).

  • Line 329- something is missing in the sentence. “…gets 5% of infants” ???

  • Line 332- how much is “small amount”? Cited study related to findings among American children 4-13 years. More importantly, I am not sure if it is suitable to compare this study’s findings with ones belong to older American children, unless there are similarities between Polish and American children and/or nurseries. This is better to be re-considered by the authors.

  • Enhancing the knowledge of nurseries staff on children’s nutritional needs and suitable beverages based on their age is important for children’s health. But the role of parents to shape children’s eating behaviours and eating patterns plus parents need for receiving essential education in this matter should not be forgotten. I suggest authors to address this at some stage through the recommendations.

  • Line 277. In discussion section the need for education and its effectiveness has been mentioned several times in different ways- it reflects a kind of repetition. Authors need to review this section.

Minor amendments

  • Table 1- Children <3 year of age not <1 year of age
  • Line 251- Table 2, the font size for the used symbols are too small to read
  • Line 251- Table 2, NS stand for Not Significant?
  • Line 381- missing a space

Author Response

We are very grateful for all the remarks, comments and suggestions to our manuscript. Please find below the Authors’ responses to each of the Reviewer’s comments. Changes made to the manuscript are marked in yellow. We hope it will make reading them easier.

Kind regards,

Authors

REVIEWER 1

Reviewer comments

The abstract needs to be reviewed for considering more clarifications e.g. line 12 “...served with meals and between meals” considering adding ‘to children’.

  • Authors’ response: It has been corrected.

Line 26- the conclusion stated in abstract must be consistent with the manuscript conclusion.

  • Authors’ response: The conclusion in the abstract is consistent with the conclusion at the end of the manuscript.

Line 135- for Figure 1 some details related to exclusion criteria e.g. full-board nutrition must be added. Generally, figures and tables must stand independent from the body of the draft.

  • Authors’ response: This detail related to exclusion criteria is shown in Figure 1 – highlighted in yellow.

There are several times that authors used “small children” instead of “young” or “toddler” or infant- this must be amended through the whole manuscript with a consistency. For example, line 151 or line 327.

  • Authors’ response: These terms have been standardized. We only used the term infant and young children.

There are few grammar errors, such as line 174- a more careful proofread is highly advised.

  • Authors’ response: The entire text has been corrected - a linguistic correction has been made.

I wonder why the number of ten-day menus and daily inventory reports has been stated differently through the body (line 194) and Figure 2 (line 199)?

  • Authors’ response: The data provided is correct. The differences are due to the fact that line 194 shows the number of ten-day menus and daily inventory reports from all analyzes (before and after education respectively from assessment I and from assessment II). The Figure 2 (line 199) shows the data separately - before and after education.

Though method section has been well written but there are different items which have not been addressed for their collection. Instead they had been referred in other sections like results or discussion sections. Examples are financial fee, types of nurseries (the definition of public and non-public has been given through the discussion, line 383), the age of participants (under 3yrs- stated in results line 233) and type of catering.

  • Authors’ response:Selected definitions such as „type of nurseries” (public vs. non-public) and „the age of children attending nurseries in Poland” are defined in the introduction, others („type of catering”, „financial fee”) in the methodology section. The repetitions have been removed from the discussion section.

 Line 236- Table 1, there is a confusion in reporting the median; min-max age- 1 (year?)- 0-43 (months?)- this must be corrected and reported in either years or months.

  • Authors’ response: The data in Table 1, concerning children attending nurseries, are expressed as number of children and percentages. The mean + SD (standard deviation), median, min - max concern the number of children not the age. The explanation is provided below the Table (highlighted in yellow). For better understanding and readability, Table 1 has been changed: the data on children and data on nurseries were separated. For better visibility and legibility, font size for symbols has been increased (similar to Table 2).

Line 232- authors stated that children under the age of 3 have been recruited but the reported mean and median for age in Table 1 for public nurseries reflecting older than that age. How this can be clarified better in defined age for recruited children?

  • Authors’ response:Nurseries, not children, were recruited for the study. Polish nurseries are attended by children under 3 years old. The data presented in Table 1 for public nurseries (what the Reviewer writes does not concern the age but the number of children). This explanation is provided in the answer to the previous question and under Table 1.

Line 236- Table 1, more information such as df and χ2 has been missed for the done Chi2 test to reveal the differences between type of nurseries.

  • Authors’ response: The data has been added under the Table 1.

Line 308- the sentence needs to be supported by a suitable reference, because nurseries feedback on given lectures through the intervention (I mean the given quote, line 309) is not related to the aim of this study either its findings.

  • Authors’ response: The reference was completed.

Line 317- lack of suitable reference. Statements must be evidence- based rather than general ones without citation(s).

  • Authors’ response: The reference was completed.

Line 329- something is missing in the sentence. “…gets 5% of infants” ???

  • Authors’ response: This sentence was corrected – now it reads: “Tea was regularly offered to 5% of infants at the age of 4 months”.

Line 332- how much is “small amount”? Cited study related to findings among American children 4-13 years. More importantly, I am not sure if it is suitable to compare this study’s findings with ones belong to older American children, unless there are similarities between Polish and American children and/or nurseries. This is better to be re-considered by the authors.

  • Authors’ response: This sentence and reference have been removed.

Enhancing the knowledge of nurseries staff on children’s nutritional needs and suitable beverages based on their age is important for children’s health. But the role of parents to shape children’s eating behaviours and eating patterns plus parents need for receiving essential education in this matter should not be forgotten. I suggest authors to address this at some stage through the recommendations.

  • Authors’ response: We fully agree with the Reviewer's suggestion that the role of parents is crucial for health as well as for shaping eating habits of children. In the Eating Healthy, Growing Healthy project the nutritional recommendations were also given to parents (indirect way). This important point has been added to the introduction (lines 46-55) and at the end of the discussion.

Line 277. In discussion section the need for education and its effectiveness has been mentioned several times in different ways- it reflects a kind of repetition. Authors need to review this section.

  • Authors’ response: The discussion section has been revised and all the repetitions were removed.

Minor amendments

Table 1- Children <3 year of age not <1 year of age

  • Authors’ response: The sentence “Children < 1 year of age” is correct. Polish nurseries are attended by children up to 3 years old, what includes infants. However, infants represent a small percentage of the total.

Line 251- Table 2, the font size for the used symbols are too small to read

  • Authors’ response: For better visibility and legibility, the font size for these symbols has been improved.

Line 251- Table 2, NS stand for Not Significant?

  • Authors’ response: Yes, an explanation of the abbreviation is under the Table 2 - marked in yellow.

Line 381- missing a space

  • Authors’ response: It has been corrected.

Reviewer 2 Report

  • The title does not reflect very well the aim of the paper-The term "water supply" is not appropriate as used here.
  • The statistical analysis used and significance P-value should be reported in abstract.
  • The current introduction does not seem to justify the importance of the work or follow a clear logical flow. I would suggest authors to rewrite the introduction to make the paper stronger. Specifically, in the introduction, the authors may want to answer the following questions:
  • Why is the work significant? Why this study needs to be examined in day-care centers/nurseries and not other types of care? What is the level of support parent participation in the labor force with regard to the context of the study? Parents in the labor force are more likely to have children attending childcare centers. Participation by parents in the labor force may lead to unhealthy eating habits among children. I would suggest authors to look at this study (International Journal of Consumer studies 2018; 42(5): 522-532).
  • What is the current knowledge gap?
  • How will this study fill in the current knowledge gap?

I've also identified minor concerns: Line 56-57: Please provide additional information about the results of these studies. Line 61-63: Please provide reference/s here and add what new evidence this study contributes (see my comments above).

  • Line 73: The study design should be clearly defined.
  • Line 136-137: repetitive- please delete.
  • Figure 1: "baseline qualitative nutritional assessment"- meaning unclear. Please clarify whether participants are interviewed in nurseries.
  • Line 143-152: It is unclear to me whether this study focused on "water supply only or on beverages in general.
  • Line 153-163: The nutrition program should be comprehensively described. Please give more information about workshops. How many dietitians/nutritionists…etc are involved?
  • Line 224-225. Why did the authors perform analysis separately for public and non-public nurseries? The authors should provide a clear reasoning here and in introduction.
  • Table 1 is unclear to me. Significant/non significant p-values should be included in a separate column.
  • Line 241-244, 260-276: Chi2 Pearson and U Mann-Whitney test, as well as P-values should be reported.
  • I would suggest adding more recent references from other contexts to discuss your results.
  • The standard of English needs to be improved throughout.

Author Response

We are very grateful for all the remarks, comments and suggestions to our manuscript. Please find below the Authors’ responses to each of the Reviewer comments. Changes made to the manuscript are marked in blue. We hope it will make reading them easier.

Kind regards,

Authors

REVIEWER 2

Reviewer comments

The title does not reflect very well the aim of the paper-The term "water supply" is not appropriate as used here.

  • Authors’ response: The title was corrected, now it reads: “Nutritional education is an effective tool in improving beverages assortment in nurseries in Poland: A nationwide cross-sectional survey”.

The statistical analysis used and significance P-value should be reported in abstract.

  • Authors’ response: The missing information about the statistical analysis used and significance P-value was reported in the abstract.

The current introduction does not seem to justify the importance of the work or follow a clear logical flow. I would suggest authors to rewrite the introduction to make the paper stronger. Specifically, in the introduction, the authors may want to answer the following questions:

Why is the work significant? Why this study needs to be examined in day-care centers/nurseries and not other types of care? What is the level of support parent participation in the labor force with regard to the context of the study? Parents in the labor force are more likely to have children attending childcare centers. Participation by parents in the labor force may lead to unhealthy eating habits among children. I would suggest authors to look at this study (International Journal of Consumer studies 2018; 42(5): 522-532).

What is the current knowledge gap?

How will this study fill in the current knowledge gap?

  • Authors’ response: The introduction was corrected in line with the reviewer's comments - the key issues raised in the review were highlighted in blue.

I've also identified minor concerns:

Line 56-57: Please provide additional information about the results of these studies.

  • Authors’ response: Additional information has been provided (lines 68-72).

Line 61-63: Please provide reference/s here and add what new evidence this study contributes (see my comments above).

  • Authors’ response: After revising the introduction, this sentence was deleted.

Line 73: The study design should be clearly defined.

  • Authors’ response: Thank you for your suggestion. We agree that it is important to understand the study design. Article was supplemented with Figure S1 presenting The overview of the project EHGH. The program description in great details has already been published in previous articles. Currently, in order to avoid repetition of the content, several facts have been quoted and referenced to what can be found in published works. Those articles are listed below and have been added to the references.

  1. Harton, A.; Myszkowska-Ryciak, J. Types of Milk and/or Its Substitutes Given to Children (6–36 Months) in Nurseries in Poland: Data from the Research and Education Project “Eating Healthy, Growing Healthy”. Int. J. Environ. Res. Public Health 2018, 15, 2789; doi:10.3390/ijerph15122789
  2. Harton, A.; Myszkowska-Ryciak, J. Nutrition Practices In Nurseries In Poland - Initial Results Of Nationwide Study. Rocz Panstw Zakl Hig. 2018, 69, 23-9.
  3. Myszkowska-Ryciak, J.; Harton A. Impact of Nutrition Education on the Compliance with Model Food Ration in 231 Preschools, Poland: Results of Eating Healthy, Growing Healthy Program. Nutrients 2018. doi: 10.3390/nu10101427.
  4. Myszkowska-Ryciak, J.; Harton, A. Eating Healthy, Growing Healthy: Impact of a Multi-Strategy Nutrition Education on the Assortments of Beverages Served in Preschools, Poland. Int J Environ Res Public Health 2018. doi: 10.3390/ijerph15071355.
  5. Myszkowska-Ryciak, J.; Harton, A. Do preschools offer healthy beverages to children? A nationwide study in Poland. Nutrients 2017. DOI: 10.3390/nu9111167

Line 136-137: repetitive- please delete.

  • Authors’ response: The repetitive fragments have been deleted.

Figure 1: "baseline qualitative nutritional assessment"- meaning unclear. Please clarify whether participants are interviewed in nurseries.

  • Authors’ response: This sentence has been corrected. The word " qualitative " has been removed. All descriptions for Figures 1 and 2 as well as the whole text of manuscript have been unified. Participants in the nurseries were not interviewed. The assessment was based on ten-day menus (10 consecutive days) and daily inventory documents. Evaluation of the nutritional education was based on the assortment of beverages served in the nurseries before and after education.

Line 143-152: It is unclear to me whether this study focused on "water supply only or on beverages in general.

  • Authors’ response: Thank you for this comment, we fully agree . Revising the manuscript we made sure that the title underlines the purpose of the study, not only (as before) the most important observations concerning only water. Hence, we changed the entire title. As stated in the abstract and in the introduction “The aim of the study was to assess the impact of education on improving the assortment of beverages served in nurseries in Poland. This analysis focused on the assortment of beverages served with meals and between meals to children. We examined: water, tea, juices, soft/fruit drinks, compote, milk, cocoa, and coffee substitute”. One of the criteria for inclusion in the study was access to drinking tap water described in methodology (previously described lines 143-152, currently 172-181). It is an important element of nurseries qualification because water supply between meals was assessed.

Line 153-163: The nutrition program should be comprehensively described. Please give more information about workshops. How many dietitians/nutritionists…etc are involved?

  • Authors’ response: The nutritional education was prepared by experts such as dieticians and nutritionists as well as behavioral scientists and healthcare practitioners. The nutritional education was conducted by 180 trained educators. The educators were people with higher education who completed their studies in the field of dietetics or human nutrition. They were specially recruited for the program EHGH and underwent a 6 months training before the program began. The whole course was completed with an exam, verification of acquired knowledge, and obtaining the EHGH Educator certificate.
  • The curriculum 12 topics included: General recommendations of balanced nutrition; Water and its role in children diet; Sugar in children diet; Prevention of food allergies; Fussy eater or food neophobic; Strong bones and teeth - the role of vitamin D and calcium; Salt in children diet; Attractive presentation of the meals; Plump, overweight, obese - how to recognize the problem; Limits of child’s choice; Involving kids in cooking; Servings size-self-eating, self-deciding. The last three classes concerned psychological aspect. During them topics such as: who makes decisions about choosing products for a child, encouraging – yes or no to independent choices as well as where and what are the limits of child`s choices (child role vs. caregiver role).
  • These facts appear in the text of manuscript, some of them have now been supplemented. Other detailed content on nutritional education in the project EHGH was included in the previously published works, which are included in the reference list.

A few selected facts have been completed. The detailed characteristics of the nutrition program were presented in the previously published articles and we referred to them in the current manuscript.

Line 224-225. Why did the authors perform analysis separately for public and non-public nurseries? The authors should provide a clear reasoning here and in introduction.

  • Authors’ response: Public and non-public nurseries in Poland differ from each other in many respects: method of financing, the number of children who attend them, the type of organization of nutrition (catering own kitchen), the money they spend on the children nutrition including beverages. These differences between public and non-public nurseries are shown in Table 1. At this point, we also wanted to check whether the nutritional education of the staff had an impact on the assortment of drinks served in nurseries and whether there were differences in this respect between public and non-public nurseries.  An explanation has been added to the introduction as suggested by the Reviewer.

Table 1 is unclear to me. Significant/non significant p-values should be included in a separate column.

  • Authors’ response: A separate column for those data has been made as suggested.

Line 241-244, 260-276: Chi2 Pearson and U Mann-Whitney test, as well as P-values should be reported.

  • Authors’ response: The missing data have been completed under the Table 1.

I would suggest adding more recent references from other contexts to discuss your results.

  • Authors’ response: New references have been added as suggested by the Reviewer.

The standard of English needs to be improved throughout.

  • Authors’ response: The writing style has been improved.

Round 2

Reviewer 2 Report

Dear Authors,

I believe that the paper has been greatly improved by these revisions. I have one comment- the study design should be clearly mentioned. Whether it is cross-sectional or longitudinal? (Line 104).

Author Response

Dear Reviewer,

Thank you very much for the positive feedback on our manuscript.

Also thank you for pointing out the type of our study design.

Indeed, our study does not meet criteria for a cross-sectional character due to the observations made at two time points. However, a longitudinal study, like a cross-sectional one, is observational (researchers do not interfere with their subjects). While we examined the effect of education (which is an intervention). Therefore, we remove the term "cross-sectional study" from the title of the article.